# Novel Anti-Neuroinflammatory Properties of a Thiosemicarbazone–Pyridylhydrazone Copper(II) Complex

**DOI:** 10.3390/ijms231810722

**Published:** 2022-09-14

**Authors:** Xin Yi Choo, Lachlan E. McInnes, Alexandra Grubman, Joanna M. Wasielewska, Irina Belaya, Emma Burrows, Hazel Quek, Jorge Cañas Martín, Sanna Loppi, Annika Sorvari, Dzhessi Rait, Andrew Powell, Clare Duncan, Jeffrey R. Liddell, Heikki Tanila, Jose M. Polo, Tarja Malm, Katja M. Kanninen, Paul S. Donnelly, Anthony R. White

**Affiliations:** 1Department of Pathology, The University of Melbourne, Parkville, VIC 3052, Australia; 2Department of Anatomy and Developmental Biology, Monash University, Clayton, VIC 3800, Australia; 3Department of Pharmacology and Therapeutics, The University of Melbourne, Parkville, VIC 3052, Australia; 4Development and Stem Cells Program, Monash Biomedicine Discovery Institute, Wellington Road, Clayton, VIC 3800, Australia; 5School of Chemistry, Bio21 Institute for Molecular Science and Biotechnology, The University of Melbourne, Parkville, VIC 3052, Australia; 6Australian Regenerative Medicine Institute, Monash University, Clayton, VIC 3800, Australia; 7Faculty of Medicine, University of Queensland, St. Lucia, QLD 4072, Australia; 8Mental Health and Neuroscience, QIMR Berghofer Medical Research Institute, Brisbane, QLD 4006, Australia; 9A.I. Virtanen Institute for Molecular Sciences, University of Eastern Finland, 70211 Kuopio, Finland; 10Florey Institute of Neuroscience and Mental Health, The University of Melbourne, Parkville, VIC 3052, Australia; 11Centre for Drug Candidate Optimisation, Monash Institute of Pharmaceutical Sciences, Monash University, Parkville, VIC 3052, Australia

**Keywords:** copper, inflammation, Alzheimer’s disease, microglia

## Abstract

Neuroinflammation has a major role in several brain disorders including Alzheimer’s disease (AD), yet at present there are no effective anti-neuroinflammatory therapeutics available. Copper(II) complexes of bis(thiosemicarbazones) (Cu^II^(gtsm) and Cu^II^(atsm)) have broad therapeutic actions in preclinical models of neurodegeneration, with Cu^II^(atsm) demonstrating beneficial outcomes on neuroinflammatory markers in vitro and in vivo. These findings suggest that copper(II) complexes could be harnessed as a new approach to modulate immune function in neurodegenerative diseases. In this study, we examined the anti-neuroinflammatory action of several low-molecular-weight, charge-neutral and lipophilic copper(II) complexes. Our analysis revealed that one compound, a thiosemicarbazone–pyridylhydrazone copper(II) complex (CuL^5^), delivered copper into cells in vitro and increased the concentration of copper in the brain in vivo. In a primary murine microglia culture, CuL^5^ was shown to decrease secretion of pro-inflammatory cytokine macrophage chemoattractant protein 1 (MCP-1) and expression of tumor necrosis factor alpha (*Tnf*), increase expression of metallothionein (*Mt1*), and modulate expression of Alzheimer’s disease-associated risk genes, *Trem2* and *Cd33*. CuL^5^ also improved the phagocytic function of microglia in vitro. In 5xFAD model AD mice, treatment with CuL^5^ led to an improved performance in a spatial working memory test, while, interestingly, increased accumulation of amyloid plaques in treated mice. These findings demonstrate that CuL^5^ can induce anti-neuroinflammatory effects in vitro and provide selective benefit in vivo. The outcomes provide further support for the development of copper-based compounds to modulate neuroinflammation in brain diseases.

## 1. Introduction

Alzheimer’s disease (AD) is a progressive neurodegenerative disorder characterised by hallmark neuropathologies including the presence of extracellular amyloid beta (A*β*) plaques, intracellular neurofibrillary tangles (NFT) composed of tau, a microtubule-associated protein, and progressive synaptic and neuronal loss. Despite large numbers of clinical trials, predominantly targeting A*β* aggregation, there currently remains no effective disease-modifying treatment for AD.

Studies have shown that neuroinflammation has a major role in the onset and progression of AD [1,2,3,4,5]. Neuroinflammation is largely mediated through microglia and astrocytes. The latter are supporting cells that help maintain key neuronal functions, including synaptic connections essential for memory and cognitive functions [6,7]. Microglia are the resident immune surveillance cell of the brain and the key effectors of neuroinflammation in AD, which represent a promising target for AD therapeutics.

Microglia are central to brain homeostasis, continuously surveying their surrounding environment with motile cell processes [8]. Upon stimulation by damaged tissue, danger signals, or infection, they alter their functional state to respond to the abnormal environment. In AD, microglia have key roles in A*β* accumulation, tau pathology, synaptic degeneration and neuron death [9]. Genetic susceptibility factors for late-onset (non-familial) AD (LOAD) have been uncovered through genome-wide association studies (GWAS) of large AD cohorts [10,11,12,13]. A significant portion of these genes (e.g., *TREM2, CR1, CD33,* and *ABCA7*) are microglia specific, or have important roles in microglial function [10,11,14,15]. Activation of pattern recognition receptors (PRRs) on microglia by aggregated A*β* leads to increased expression and release of inflammatory cytokines and chemokines including interleukins 1 (IL-1), IL-6, monocyte chemoattractant protein 1 (MCP-1) and tumour necrosis factor alpha (TNFα) [16]. Microglia are also activated in AD brain by other inflammatory molecules including interferon gamma (IFNγ). Additionally, the alteration to microglia in AD can also involve functional changes to these phagocytic cells. Phagocytosis is the uptake and degradation of unwanted material such as aggregated A*β* and dead cells and is critical for healthy brain function. Microglial phagocytosis is impaired in AD, with reduced amyloid clearance potentially associated with changes in risk gene expression including *TREM2* and *CD33* [17,18,19]. Improving microglial phagocytic activity is thus a potential therapeutic target for AD [17]. With accumulating evidence suggesting active contribution of microglia and neuroinflammation to AD pathogenesis, the development of microglia-targeting immune-modulating compounds for neurodegenerative diseases have gained research interest [20,21,22].

Deregulated biometal homeostasis is an established feature of neurodegenerative diseases. To restore biometal homeostasis, metal-regulating complexes have been investigated for therapeutic potential in neurodegenerative diseases. Membrane-permeant and stable *bis*(thiosemicarbazonato)copper(II) complexes (CuBTSCs) were examined in preclinical models of AD, amyotrophic lateral sclerosis (ALS), and Parkinson’s disease (PD)—neurodegenerative diseases with established roles for neuroinflammation [2,23,24,25,26,27,28,29]. The copper(II) complex, glyoxalbis(N(4)–methylthiosemicarbazonato)copper(II) (Cu^II^(gtsm)) was shown to exert multiple therapeutic effects in the APP^swe^/PS1^δE9^ mouse model of AD [25]. A structurally related compound, diacetylbis(N(4)-methylthiosemicarbazonato) copper(II) (Cu^II^(atsm)), lacked similar efficacy in an initial assessment in the same AD model, but had protective effects in multiple animal models of familial ALS and PD [24,26,27,28,29]. The neuroprotective and therapeutic efficacy of Cu^II^(atsm) in various models of neurodegenerative disease has been attributed to several possible modes of action including the selective delivery of copper to metal deficient superoxide dismutase and other cuproproteins in cells with impaired mitochondria and inhibition of toxicity driven by peroxynitrite [28,29,30]. Others reported treatment with Cu^II^(atsm) being protective against ferroptotic cell death by inhibiting lipid peroxidation possibly by serving as a radical trapping agent [31]. The promising preclinical work in multiple animal models of ALS and PD stimulated first-in-human trials where Cu^II^(atsm) was evaluated in both ALS (NCT02870634, NCT04082832) and PD patients (NCT03204929) [32,33].

Notably, treatment with Cu^II^(atsm) modulates the inflammatory response both in vitro [34] and in vivo [24,26,28,34,35]. Specifically, Cu^II^(atsm) has been demonstrated to attenuate astrocyte and microglial activation [24,26,28,34]. Furthermore, upregulation of metallothionein 1 (*Mt1*), a metal-regulated inflammatory response protein, by Cu^II^(atsm) supports an intimate relationship between copper-based drugs and modulation of inflammation [34,36,37,38]. These findings suggest that metal complexes may be used to concurrently target multiple disease pathologies, possibly by exerting therapeutic effects via regulation of inflammatory responses and/or metal homeostasis.

In this study, we sought to discover new immune-modulating copper(II) complexes habouring microglia-targeting properties for potential application for AD. Copper(II) complexes will be evaluated across a panel of assays using in vitro model of microglia. The copper(II) complexes were selected as each being stable, four coordinate complexes with tetradentate dianionic ligands providing overall charge-neutral lipophilic complexes with the potential to cross cell membranes and the blood–brain barrier.

## 2. Results

Five different copper(II) complexes, CuL^1−5^, were assessed in this work (Figure 1 and Figure 2a). One of the five complexes, CuL^1^, is a derivative of Cu^II^(atsm) that has been made water soluble by the addition of an aromatic sulfonate functional group [39]. The other four complexes each involve a dianionic quadridentate ligand to form stable, charge-neutral and lipophilic complexes with copper(II). Two complexes, CuL^3^ and CuL^4^, are well-known complexes of tetradentate Schiff-based ligands derived from the condensation of 1,2-diaminoethane with either of salicaldehyde (CuL^3^ or Cu(salen)), or 2-ethoxybenzaldehyde (CuL^4^). The final two complexes, CuL^2^ and CuL^5^, feature a tetradentate thiosemicarbazone-hydrazonepyridine ligand [40,41]. In the case of CuL^5^, the hydrazonepyridine ligand contains a substituted benzofuran functional group designed to enhance interaction with amyloid-β plaques [41]. However, little was known about the cell permeability of each complex prior to this study.

### 2.1. Cellular Toxicity and Copper Uptake of Copper(II) Complexes

A preliminary assessment of CuL^1−5^ was performed in a neuroblastoma cell line BE(2)-M17 (M17) to examine the toxicity of the complexes to cells and measure the stability and cell membrane permeability of each complex by measuring cellular copper concentrations following treatment. Each complex was compared to Cu^II^(atsm), a previously evaluated copper(II) complex with low toxicity both in vitro [34] and in vivo [24,26,27,28,34]. The cytotoxicity of the complexes was assessed using both an LDH cytotoxicity assay (Figure 2b) and an MTT reduction assay (Appendix A) which measures cell redox state rather than cell lysis. Treatment of M17 neuroblastoma cells (concentrations up to 10 μM) with CuL^1^, CuL^3^ and CuL^4^ did not significantly impact cell health as measured by both assays. Treatment with CuL^5^ did not induce a significant increase in released LDH but did induce a dose-dependent decrease in MTT reduction (Figure 2b and Appendix A). The effect of CuL^5^ on MTT reduction was analogous to Cu^II^(atsm). In contrast, treatment with CuL^2^ at all tested concentrations demonstrated cytotoxicity, thus was eliminated from further screening. To assess the ability of the complexes to increase cell-associated copper, total copper concentrations were measured using inductively coupled plasma mass spectrometry (ICP–MS) (Figure 2c). Incubation with CuL^1^, CuL^3^ and CuL^4^ (2.5–10 μM) resulted in no increase in cellular copper levels, indicative of either poor cell-membrane permeability or poor stability of the complexes in cell media. In contrast, incubation with CuL^5^, dose dependently increased cellular copper.

### 2.2. CuL^5^ Inhibited MCP-1 Secretion by Murine Microglia

Each complex, excluding the cytotoxic CuL^2^, was screened for its ability to modulate inflammatory activity in an in vitro neuroinflammation model [34]. Primary murine microglia cultures were stimulated using the cytokines TNFα and IFNγ (TNF/IFN) to induce a neurodegeneration-like inflammatory response as previously described [34]. The level of MCP-1 (Figure 2d), a pro-inflammatory cytokine secreted by microglia mediated through TNF/IFN, was measured using ELISA [42]. Minocycline, a drug that exerts anti-inflammatory effects in models of neurodegenerative diseases (albeit without clinical translation to date) [43,44], was used as a positive control. Concurrent stimulation and treatment of microglia with TNF/IFN and minocycline significantly reduced secretion of MCP-1 (by ~33%; Figure 2d) compared to treatment with TNF/IFN alone. Treatment of TNF/IFN-stimulated microglia with CuL^1^, CuL^3^ or CuL^4^ did not modulate the level of microglia-secreted MCP-1 (Figure 2d) but CuL^5^ induced a significant dose-dependent reduction in microglia-secreted MCP-1 at treatment concentrations of 1 and 2 μM (25% at 1 μM and 38% at 2 μM; Figure 2d) when compared to treatment with TNF/IFN alone. Treatment with concentrations of 2 μM CuL^5^ induced an effect comparable to minocycline at 20 μM.

### 2.3. CuL^5^ Increased Phagocytic Activity of Microglia

An important functional role of microglia is the removal of unwanted material including A*β* and dead cells. Multiple studies, mostly in preclinical models, have suggested impaired microglial phagocytosis in AD; hence, improving phagocytotic function may be a promising therapeutic strategy [17,45,46]. To assess if treatment of microglia with CuL^5^ would improve microglial phagocytosis, the ability of microglia to phagocytose fluorescent pHrodo red *E.coli* bioparticles was investigated. Stimulation of microglia with TNF/IFN (Figure 3(aiii)) reduced microglial phagocytosis to a level similar to treatment with cytochalasin D (Figure 3(aiv)), a known inhibitor of phagocytosis [47]. Concurrent treatment with CuL^5^ reversed the effect of TNF/IFN, inducing a dose-dependent increase in microglial phagocytosis (Figure 3(av),(avi)).

### 2.4. CuL^5^ Inhibited Expression of the Pro-Inflammatory Cytokine, TNFa and Upregulated Metallothionein

CuL^5^-mediated modulation of inflammatory responses at the transcriptional level resulted in changes in mRNA expression for pro-inflammatory cytokines. Expression of *Tnf,* encoding the pro-inflammatory cytokine TNFα, was enhanced in the presence of TNF/IFN. Addition of minocycline resulted in a trend to reduced *Tnf* mRNA expression (Figure 3(bi)). Addition of CuL^5^ concurrently to TNF/IFN-stimulated primary microglia led to a dose-dependent reduction in *Tnf* mRNA expression consistent with the inflammation-modulating effect measured by ELISA (Figure 2d and Figure 3(bi)).

Metallothioneins (MTs) are a family of metal-binding proteins that play a role in regulating homeostasis of biometals, predominantly copper and zinc (reviewed in [48]) with MT1 and MT2 being the most widely expressed isoforms in the brain (reviewed in [37]). MTs are important regulators of inflammatory responses in neuropathological conditions [36,38]. Although the expression level of *Mt1* did not change with stimulation of microglia with TNF/IFN, concurrent treatment with CuL^5^ effected a significant dose-dependent increase in *Mt1* mRNA expression (Figure 3(bi)).

### 2.5. CuL^5^ Modulated Expression of Late Onset AD risk Genes, Cd33 and Trem2

GWAS have identified several risk gene variants for LOAD, and the majority are either immune related or display microglia-restricted expression [10,12,13,14,15]. We examined the two most commonly studied microglia-associated phagocytosis inhibiting and phagocytosis promoting LOAD-risk genes of AD (after apolipoprotein E), *Cd33* [49] and *Trem2* [50] respectively, and tested if CuL^5^ transcriptionally regulates these risk factors. Following stimulation of microglia with TNF/IFN, *Trem2* was significantly reduced (Figure 3(biii)) and *Cd33* expression showed a trend towards elevation (Figure 3(biv)). While minocycline did not reverse the change in expression of either *Cd33* or *Trem2* induced by TNF/IFN, treatment with CuL^5^ reversed the TNF/IFN-induced change in expression of both *Cd33* and *Trem2* (Figure 3(biii),(biv)), suggesting that CuL^5^ may have the potential to normalize some aberrant immune-associated risk gene expression.

### 2.6. CuL^5^ Was Well Tolerated by Mice and Increased Copper Content in the Brain

The in vivo toxicity of CuL^5^ was evaluated in a dose-escalation trial (3 mg/kg–30 mg/kg over 7 days). Potential adverse drug effects were assessed through daily monitoring of the mice, and biochemical analysis of blood serum samples collected from individual animals at experimental end-point. During the trial, the mice undergoing treatment were weighed daily and no weight loss was observed in either standard suspension vehicle (SSV) (*n* = 4) or CuL^5^ (*n* = 3) treatment groups (reproduced in 8 weeks treatment (Appendix A). The serum levels of protein markers associated with liver and kidney function and health including albumin, globulins, alanine aminotransferase (ALT), aspartate aminotransferase (AST) and creatine kinase (CK) (Appendix A), urea (Appendix A) and cholesterol (Appendix A) remain largely unchanged; suggesting that CuL^5^ did not induce significant systemic adverse effects in the animals. The total copper content in brain tissue from samples collected from non-transgenic WT mice following a week of daily oral gavage with either SSV or CuL^5^ (30 mg/kg) (Figure 4a) was determined using ICP–MS. The CuL^5^ treatment group (*n* = 10) had an approximate 10% increase in brain copper content relative to the SSV treatment group (*n* = 5) (Figure 4b, *p* < 0.01).

### 2.7. CuL^5^ Treatment of 5xFAD Mice Increased Hippocampal Amyloid Load

5xFAD mice and non-transgenic littermates were treated by oral gavage with SSV or CuL^5^ daily for 8 weeks (Figure 4a). Through the treatment period, mice subjected to oral gavage from 10 weeks of age gradually gained weight and there were no significant differences in the weights of the mice across treatment groups (Appendix A), suggesting that the compounds were well tolerated. After 8 weeks of treatment with CuL^5^, mice were injected i.p. with Me-X04, a Congo-red-derived fluorescent probe, to stain for in vivo fibrillar and plaque A*β* to assess CuL^5^ effect on microglial amyloid phagocytosis. Isolation of microglia (CD11b+, CD45^mid^, CX3CR1+), revealed a trend toward an increase in the proportion of plaque phagocytosing (X04^+^) microglia following treatment with CuL^5^ (Figure 4c,d). Brain tissue samples collected from the animals following the 8 weeks treatment with CuL^5^ were examined for A*β* plaque load by immunohistochemistry (Figure 4e). We assessed the hippocampus and observed a significant increment in both the amyloid plaque numbers (Figure 4f) and percentage area (Figure 4g) of amyloid burden following treatment with CuL^5^. While this contrasts with the common aim of plaque removal, the relationship between plaque load and cognitive performance in AD is still uncertain (reviewed in [51]). As such, we further investigated if CuL^5^ could confer positive changes in cognitive performance of AD mice.

### 2.8. CuL^5^ Induced Selective Improvement of the Cognitive Function of 5xFAD Mice

5xFAD mice and their non-transgenic littermates, orally gavaged with SSV or CuL^5^ daily for up to 9 weeks, were subjected to a battery of behavioural tests (Figure 5a). Mice were assessed for long-term spatial memory in the Morris Water Maze. During the acquisition phase, no overall difference in the latency to reach the platform was seen between WT and 5xFAD mice (Figure 5b). However, on the first day of training, treatment with CuL^5^ reduced latency to reach the platform (Figure 5b: two-way ANOVA for repeated measures (ANOVA-RM): day*drug interaction; Latency: F_4,82_ = 4.7, *p* = 0.002; pairwise SVV vs. CuL^5^
*p* = 0.007) probably due to increased swim velocity (data not shown; F_4,92_ = 4.5, *p* = 0.002). In the probe trial for search bias on the 5th day when the platform was removed, all mice regardless of genotype and treatment showed preference for the previous platform location, indicating intact long-term memory of the platform (Figure 5c: ANOVA-RM: pool quadrant, F_3,69_ = 8.3, *p* < 0.001). Nest building ability was scored in mice every 2 weeks from baseline to 12 weeks post-treatment. Regardless of treatment, 5xFAD mice were impaired at nest building (Figure 5d: Ordered logistic regression: genotype effect: OR: 0.32, 95%CI: 0.15, 0.67, *p* = 0.003). All mice improved their nest building over time, and this was not modulated by genotype or treatment (Data not shown: Ordered logistic regression: time effect: OR: 1.13, 95% CI:1.07, 1.19, *p* < 0.001). Mice were also assessed for spatial working memory using the igloo test over 5 days. No difference between test days or a main effect of drug treatment was found. However, 5xFAD mice made fewer correct choices compared than WT mice overall. There was also a genotype–drug interaction, such that CuL^5^ treatment improved choice accuracy of 5xFAD mice more than of WT mice (Figure 5e: ANOVA-RM: genotype–drug interaction: F_1,23_ = 8.5, *p* = 0.008; pairwise 5xFAD-SSV vs. 5xFAD-CuL^5^
*p* = 0.009). 5xFAD mice made more perseverative errors (series of incorrect choices) than WT mice, regardless of treatment (Figure 5f: poisson regression: genotype effect: IRR: 2.07, 95% CI: 1.49, 2.88; *p* < 0.001), where CuL^5^ treatment induced a significant reduction in perseverative errors in 5xFAD mice (F_1,23_ = 4.2, *p* = 0.051; pairwise 5xFAD-SSV vs. 5xFAD-CuL5, *p* = 0.02) (Figure 5f).

## 3. Discussion

Despite several decades of research, there remains no effective treatment to halt or slow down the progression of AD. Growing evidence supports a major role for altered immune system function in AD and, in particular, changes to microglial function [52,53,54,55]. Microglia are also considered an important therapeutic target for inflammation modulating compounds. Previous studies on the copper(II) complex, Cu^II^(atsm), revealed the potential for copper-based drugs to modulate neuroinflammation in degenerative brain diseases [24,25,26,28]. In this study, we investigated a small family of five related copper(II) complexes to determine their potential to modulate neuroinflammatory responses in vitro and in vivo.

The primary basis for testing the inflammation-modulating activity of the new copper(II) complexes was the discovery of anti-inflammatory activity exerted by Cu^II^(atsm) in SOD1G93A and SOD1G37R mouse models of ALS as well as in a preclinical mouse model of ischemic stroke [24,26,35]. Since findings suggest that copper(II) complexes possess potential multi-targeting properties for the treatment of neurodegeneration, we aimed to screen different complexes as candidate(s) for exerting effects on inflammatory pathways. Chronic neuroinflammation is a widely established pathology of AD, characterised by the encircling of Aβ plaques by microglia and reactive astrocytes. Microglia, the specialized immune sentinel cells of the brain, serve important roles in regulating inflammatory responses in the CNS, so primary murine microglia cultures were selected as a suitable model for mimicking neuroinflammation in vitro. To achieve a physiologically relevant model of neuroinflammation in vitro, cytokines associated with AD pathogenesis (reviewed in [56,57]) were used to stimulate microglia. We found that the copper(II) complex CuL^5^, a thiosemicarbazone–pyridylhydrazone that incorporates a benzofuran functional group, reduced secretion of pro-inflammatory cytokines MCP-1, decreased expression of *Tnf*, increased expression of *Mt1*, and modulated expression of Alzheimer’s disease-associated risk genes, *Trem2* and *Cd33* in murine microglia. CuL^5^ also improved the phagocytic function of microglia in vitro. In vivo, treatment of 5xFAD mice with CuL^5^ for 8 weeks led to improved performance in the igloo test for working memory. These findings demonstrate that the thiosemicarbazone–pyridylhydrazone, CuL^5^ can induce anti-neuroinflammatory outcomes in vitro and provide selective benefit in vivo.

### 3.1. Inflammation-Modulating Efficacy of CuL^5^

In AD, microglia are a primary source of inflammatory cytokines (reviewed in [16]), involved in mediating neurotoxic effects [58], but also participate in the clearance of Aβ plaques through phagocytosis [17,59]. Due to the multi-functional capacity of microglia, it remains debatable if the role of microglia is beneficial, damaging or a mixture of both in AD [59]. In this study, we showed that treatment of microglia with CuL^5^ stimulated with TNF/IFN can dampen microglial secretion of MCP-1, suggesting an inflammation-modulating effect of the compound. Subsequently, we were also interested to determine if treatment with CuL^5^ mediated phagocytosis-associated changes in microglia. Changes in microglial expression of immune receptors, TREM2 and CD33, which are implicated as risk variant genes in LOAD [10,12,13], can alter the phagocytic capacity of microglia [18,19]. Specifically, CD33 triggers the inhibition of phagocytosis while TREM2 signalling stimulates phagocytosis and altered expression levels of CD33 and TREM2 have been reported in samples from AD patients [60,61,62,63]. Stimulation of microglia with TNF/IFN was associated with a trend towards an increase in *Cd33* mRNA expression level and a reduction in *Trem2* mRNA expression level, suggesting that stimulation of microglia with TNF/IFN could potentially reduce the phagocytic capacity of microglia cells, supporting in vivo data. When CuL^5^ was added concurrently with TNF/IFN, the effects of TNF/IFN on mRNA expression levels of *Cd33* and *Trem2* were reversed. This was also consistent with our observation of improved phagocytic activity. Additionally, while the in vivo states of microglia are diverse (reviewed in [64]), it was observed in previous studies that transcriptomic profiles of disease-associated microglia [65] and amyloid phagocytosing microglia [59] from 5xFAD mice reflect an increased *Trem2* expression and *Cd33* downregulation. Our findings suggest that the delivery of CuL^5^ into the brain could potentially confer beneficial outcomes in AD through increasing microglial phagocytosis of Aβ.

### 3.2. CuL^5^-Mediated Anti-Inflammatory Action May Be Linked to MT

Metallothioneins (MTs) are rapidly upregulated in response to inflammation, hence their designation as acute phase proteins. Inflammation may activate MT expression through multiple pathways, including directly by stimulating an antioxidant response element and specific metal response elements in the promoter region or indirectly by second-messenger protein kinase pathways [66]. In the CNS, MT1 and MT2 are predominantly expressed by astrocytes, and expression by microglia is induced in reactive but not resting microglia (reviewed in [37]). In AD, potential disease-modulating mechanisms of MTs have been demonstrated both in vitro and in vivo. In vitro, astrocyte-secreted MT1 was demonstrated to reverse Aβ-induced cytotoxicity in N2a neuroblastoma cells and attenuated Aβ-induced microglia neurotoxicity in BV-2 microglia cells [67]. In vivo, crossing *Mt1*-overexpressing mice with the APPTg2576 mouse model of AD resulted in varied outcomes [68]. Since MT is involved in the scavenging of reactive oxygen species (ROS) and reactive nitrogen species [69], the upregulation of MT in AD could potentially be a protective mechanism against AD-associated inflammation and oxidative damage. In this work, we observed that CuL^5^ induced a dose-dependent increase in *Mt1* mRNA expression in microglia. It has been reported previously that MT1 can modulate MCP-1 expression, and we have shown that Cu^II^(atsm) was also able to significantly reduce MCP-1, IL-6 and and TNFa in primary microglia and astrocyte cultures, together with elevated MT1 [34,70]. Whether these are directly related, or parallel phenomena, remains to be determined, but our work does highlight consistent effects of cell-permeant copper(II) complexes on MT1 and decreased inflammatory markers in microglia. There is also a potential role for CuL^5^-mediated MT1 changes in the altered *Trem2* and *Cd33* gene expression. CuL^5^ increased expression of *Trem2* and decreased *Cd33*. There are no reports of MT-mediated actions on either of these AD risk genes, thus further research is warranted. Our findings are however unique, in showing that a copper(II) complex can normalize the expression of those two significant AD risk genes. CuL^5^-induced upregulation of *Mt1* mRNA expression could potentially confer a protective effect in AD, although this remains to be investigated in vivo. Alternatively, upregulation of *Mt1* could confer different effects in different microglia subsets [59,65,71]. A single-cell approach could be used to assess drug treatment effects on heterogenous cell populations such as microglia.

### 3.3. Paradoxically, CuL^5^ Increases Phagocytic Function In Vitro but Induces Accumulated Amyloid In Vivo

Microglial phagocytosis in AD is considered to be impaired [45,46]. Although microglia become activated during the disease, they are known to be defective in their ability to phagocytose and/or digest aggregated amyloid in the brain [17,72]. Therefore, a well-accepted therapeutic approach is to seek drugs that improve microglial phagocytic activity. Here, we show that CuL^5^ promotes increased phagocytic function in primary murine microglia. Microglia treated with TNF/IFN showed a reduction in phagocytosis of bioparticles, while concurrent treatment with CuL^5^ significantly increased phagocytosis. Whether this is directly related to the changes observed in *Trem2* and/or *Cd33* expression induced by CuL^5^ is unknown; however, the relationship between *Trem2* and *Cd33* expression and microglial phagocytic activity have been previously reported [18,19]. To confirm the role of *Trem2* in CuL^5^ action on phagocytosis, suppression of *Trem2* expression should be investigated. Interestingly, copper can reduce phagocytic function, likely due to exposure to toxic levels of the metal [73]. In contrast, other reports have demonstrated improved phagocytosis in peripheral (non-CNS) macrophages by copper(II) complexes, including copper acyl salicylate, and polypyridyl ligand [74,75]. However, here we report the first observation of copper(II) complex-mediated increase in microglial phagocytosis. Together with the previous reports on copper complex effects on macrophages, our work supports the further need to investigate copper-based drugs as promoters of phagocytosis in various macrophage populations.

Although CuL^5^ induced phagocytosis in cultured murine microglia, we found that, in vivo, the treatment increased the area of compact amyloid plaques in the AD model mice. It is uncertain whether this is related to altered phagocytic activity in the CuL^5^-treated mice, or to other factors controlling amyloid accumulation and organization into compact plaques. In many transgenic mouse models of AD carrying familial AD linked mutations that drive the overproduction of Aβ, increased amyloid burden has been found to correlate with impaired memory (reviewed in [76]). As such, therapeutic candidates that reduce Aβ plaque load often demonstrate beneficial effects on cognitive outcomes in transgenic AD mice, albeit without translating to clinical improvement in AD patients [77,78,79,80,81,82,83]. However, it is increasingly evident that soluble oligomeric Aβ is a more likely pathogenic agent in AD (reviewed in [84]). Specifically, oligomeric species of Aβ isolated from human AD brain tissues or generated in vitro have been shown to impair cognitive performance when administered to rodents [85,86,87,88,89]. Given the equilibrium between dense core plaques and oligomers, it is possible that CuL^5^ decreased the tissue levels of toxic Aβ oligomers by augmenting the formation of compact amyloid plaques. There is evidence that amyloid plaques exert binding capacity to soluble amyloid species [90]. To additionally investigate the role of CuL^5^ in influencing toxic Aβ species, the levels of oligomeric Aβ in brain tissues collected from animals that received treatment should be assessed in future studies [91].

Interestingly, Aβ42 contains also a Cu^2+^ binding site that promotes the precipitation and aggregation of Aβ42 peptides in the presence of Cu^2+^ [92]. Resulting from its high binding affinity for Cu^2+^, amyloid plaques may be acting as copper ‘sinks’, leading to copper deficiency in the surrounding environment. As substituted benzofurans with high affinity for amyloid plaques have been used as diagnostic imaging agents, incorporation of the benzofuran-5-ol motif into the organic backbone of CuL^5^ was aimed at targeting Aβ plaques, to improve delivery of the complex to the affected regions of the brain (plaques). Following dissociation from the L5 ligand backbone, Cu^2+^ delivered to the target brain region could be captured by Aβ42 peptides, consequently leading to accelerated conversion of Aβ oligomers to the ‘safer’ highly aggregated form, a therapeutic strategy also adopted by Zhang and colleagues. This will however require further validation. If validated, it would indicate the feasibility of directed compound delivery through modification of chemical groups on the organic ligand. This work also suggests that a potentially better therapeutic approach than targeting to plaques, would be to design compounds that specifically target microglia (e.g., X04+ microglia) surrounding the plaques. One such approach could involve taking advantage of cell surface markers with upregulated expression in X04+ microglia (e.g., TREM2, APOE, CD115) [59].

### 3.4. CuL^5^ Improves Short-Term Memory Function in AD Model Mice

To determine if CuL^5^ induced therapeutic benefit, 5xFAD and wild-type mice were treated by oral gavage and subjected to three different measures of learning and memory, including MWM, nest building, and igloo test. CuL^5^ had no significant effect overall on long-term memory and planning in 5xFAD or WT mice, as measured by the MWM and nest building tests, respectively. The lack of efficacy in the MWM could indicate a lack of effect of CuL^5^ on hippocampal-dependent long-term memory [93,94]. The failure to modify nest building indicates no benefit of the compound on a complex species-specific behaviour involving hippocampus- and prefrontal cortex-dependent planning of actions [95,96,97,98,99]. The small but significant improvement in the proportion of correct choices of the igloo test by 5xFAD mice treated with CuL^5^ could suggest that the compound can specifically improve short-term spatial working memory. The reason for such specific benefit of CuL^5^ in the one test may be related to reliance on different neural circuits (hippocampus vs. medial frontal cortex) and memory mechanisms (long-term vs. short-term) in the tasks. As previously mentioned, CuL^5^-treatment increased plaque accumulation in 5xFAD mice, contrary to our hypothesis [25]. In the present study, increased plaques in 5xFAD and WT mice did not lead to decreased memory function more broadly, consistent with some animal studies showing a disconnect between amyloid levels and memory deficits, a finding that is also observed in some studies on patients with AD [51,100,101,102]. Further studies incorporating in ex. battery of short-term learning, motivation and anxiety tests as well as different duration of CuL^5^ treatment will be needed to determine whether other functional tests of behaviour determine additional specific benefits of treatment with copper(II) complexes.

### 3.5. Summary of Findings, Limitations and Implications

Neuroinflammation plays a fundamental, albeit, poorly understood, role in AD pathogenesis. However, available treatment options targeting neuroinflammation are inadequate, providing limited efficacy. We have previously demonstrated copper(II) complex delivery as a potential approach for modulating neuroinflammatory conditions. In this study, we report the first observation of copper(II) complex-mediated increase in microglia phagocytosis coupled with selective benefit in vivo. This study demonstrates the multi-targeting effects of copper(II) complexes hence their potential as candidates in the development of therapeutics for complex diseases.

Nonetheless, we have mainly focused on using a microglial model, the primary driver of neuroinflammation, during the preliminary screening of copper(II) complexes. We demonstrated that copper(II) complexes can act on microglia to confer beneficial effects potentially via amyloid phagocytosis. However, neuroinflammation is a tightly regulated process that involves the complex interplay of many cell types. This study lacks information regarding the direct impact of CuL*^5^* on other cell types including neuronal cells, the key producers of amyloid and phosphorylated tau. The link between increased plaque load with CuL*^5^* treatment and its potential implication are unclear and hence will require further evaluation in future studies.

## 4. Materials and Methods

### 4.1. Neuroblastoma Cell Line BE(2)-M17

BE(2)-M17 neuroblastoma cells were cultured in a T-75 cm^2^ flasks (Thermo Scientific, Waltham, MA, USA) in Opti-MEM (Life Technologies, Carlsbad, CA, USA) supplemented with 1X nonessential amino acids (NEAA; Life Technologies, Carlsbad, CA, USA), 1X sodium pyruvate (Life Technologies, Carlsbad, CA, USA), 10 % (*v*/*v*) fetal bovine serum (FBS) (Bovogen, Keilor East, VIC, Australia) and 50 U/mL penicillin and 50 µg/mL streptomycin (Life technologies). Cells were detached using 1× trypsin EDTA solution (Sigma-Aldrich, Burlington, MI, USA) in phosphate buffered saline (PBS) (Life Technologies, Carlsbad, CA, USA) and seeded at 3 × 10^5^ cells/mL concentration in 12- (1 mL) and 24- (0.5 mL) well culture plates (Thermo Scientific, Waltham, MA, USA) for experiments. Cells were maintained in a humidified incubator at 37 °C with 5% CO_2_.

### 4.2. Primary Cell Culture

#### 4.2.1. Animals

All use and handling of animals for experimentation were approved by University of Melbourne Animal Experimentation Ethics Committee (SEC no. 1312831), Monash Animal Ethics Committee (MARP-2016-112) and Animal Experiment Committee in State Provincial Office of Southern Finland (Licence no. ESAVI-2018-012856), and conformed with national and institutional guidelines. Animals were housed in specific pathogen-free conditions with 12 h light/dark cycle with water and food available ad libitum, unless stated otherwise. Newborn pups from time-mated C57BL/6J were taken between days 0–1 for primary mixed glial culture. Day 0–1 pups were anaesthetized by inducing hypothermia. Heterozygous 5xFAD transgenic mice on a B6SJL hybrid background were used for oral gavage trials. Non-transgenic littermates were concurrently used for experimental controls. Animals did not undergo any prior procedures before they were used for gavage trials. At 10 weeks of age, sex-matched 5xFAD and non-transgenic animals were subjected to oral treatment with compounds, as indicated. Animals subjected to oral gavage were fasted for 4 h prior to the procedure.

#### 4.2.2. Primary Mixed Glial Culture

Mixed glial cultures were harvested as described in [34], originally described by [103]. Brains were removed from decollated neonatal wild-type (WT) mice and submerged in ice-cold preparation buffer (0.137 M NaCl, 5.35 mM KCl, 0.22 mM KH_2_PO_4_, 0.17 mM Na_2_HPO_4_, 5.55 mM glucose, 58.5 mM sucrose, 200 U/mL penicillin and 200 µg/mL streptomycin). Brains were then sequentially strained through mesh sized 250 μm and 135 μm, collected in ice-cold preparation buffer and centrifuged at 500× *g* for 5 min at room temperature (RT). The cell pellet was resuspended in mixed glial growth media (high glucose DMEM; Life Technologies, Carlsbad, CA, USA) supplemented with 10% (*v*/*v*) FBS, 50 U/mL penicillin and 50 µg/mL streptomycin (Life Technologies, Carlsbad, CA, USA). Cells were seeded at 3 × 10^5^ cells/mL in T-175 cm^2^ flask (Thermo Scientific, Waltham, MA, USA) and maintained in a humidified incubator at 37 °C with 10% CO_2_. Mixed glial growth media were replaced every 7 days after plating. Cultures were used for primary microglia harvest upon maturation for 17 days in vitro (DIV).

#### 4.2.3. Primary Microglia Culture

Microglia were harvested from mixed glial cultures on 17 DIV by mild trypsinisation, as described in [34]. Conditioned media were collected and filtered through a 0.22 μm bottle top filter (Corning Inc., Corning, NY, USA). For the trypsinisation and washing steps, 20 mL of appropriate media were added to T175 flasks, unless otherwise stated. Mixed glial cultures were rinsed with high glucose DMEM followed by 20-25 min trypsinisation with trypsin EDTA (1.67× Trypsin EDTA solution in high glucose DMEM), until complete detachment of the astrocyte layer. Adherent microglia were scraped into 4 mL/flask of conditioned media, collected and centrifuged at 405× *g* for 10 min (RT). The resultant microglial pellets were pooled together, and resuspended in conditioned media. Microglia were seeded at 1.25 × 10^5^ cells/mL in 12- (800 μL), 24- (400 μL) and 96- (100 μL) well culture plates (Thermo Scientific, Waltham, MA, USA). Microglia were maintained in conditioned media at 37 °C with 10% CO_2_. After resting overnight, conditioned media were replaced with microglial growth media (IMDM (Life Technologies, Carlsbad, CA, USA) supplemented with 10% (*v*/*v*) FBS, 2mM L-glutamate (Life Technologies, Carlsbad, CA, USA), 50 U/mL penicillin and 50 µg/mL streptomycin) and incubated at 37 °C with 5% CO_2_ for 72 h prior to experiments.

### 4.3. Copper(II) Complexes Synthesis

Copper(II) complexes were synthesised as described previously:Cu^II^(gtsm) [40], Cu^II^(atsm) [104],CuL^1^, Sodium diacetyl-4-methyl-*p*-benzenesulfonato-bis(thiosemicarbazonato)copper(II) [39],CuL^2^, Diacetyl-2-pyridylhydrazone-N4-methyl-3-thiosemicarbazone [40],CuL^3^, *N*,*N*′-ethylenebis(salicylaldimine)copper(II) [105], andCuL^4^, *N*,*N*′-ethylenebis(3-ethoxysalicylaldimine)copper(II).H_2_O [106].

#### 4.3.1. 2-Hydrazino-pyridinyl-5-(5′-methoxybenzofuran) (**1**)

To a flask charged with 2-chloro-pyridinyl-5-(5′-methoxybenzofuran)([41]) (153 mg, 0.59 mmol) was added hydrazine hydrate (3 mL, 65%). The mixture was then heated to reflux for 2 h under and atmosphere of N_2_. During this time the remaining solid had dissolved to give an orange colour and subsequent heating resulted in the precipitation of a beige solid. The reaction mixture was then allowed to cool and the solid was collected by filtration, washed with a copious amount of water followed by ether to yield a beige powder (74 mg, 0.29 mmol, 49%). ESI–MS: HPLC: ^1^H NMR (500 MHz; d_6_-DMSO) δ 8.52 (1H, s, ArH), 7.91 (2H, m, ArH+NH), 7.45 (1H, d, ArH, *J* = 8.9 Hz), 7.07 (2H, m, ArH), 6.82 (2H, m, ArH), 4.28 (2H, d, NH_2_, *J* = 0.5 Hz), and 3.78 (3H, s, CH_3_, *J* = 1.1 Hz). ^13^C NMR (126 MHz, d_6_-DMSO) δ 161.7, 155.7, 155.3, 148.7, 144.4, 133.4, 129.8, 114.9, 111.8, 111.2, 106.0, 103.1, 99.0, and 55.5.

#### 4.3.2. Diacetyl-2-(2-hydrazino-pyridinyl-5-(5′-methoxybenzofuran))-(4-methyl-3-thiosemicarbazone) (H_2_L^5^)

A suspension of 2-hydrazino-pyridinyl-5-(5′-methoxybenzofuran) (125.1 mg, 0.49 mmol) and diacetyl-*mono*-4-methyl-3-thiosemicarbazone [107] (86.0 mg, 0.50 mmol) in ethanol (25 mL) was acidified with a few drops of acetic acid. The mixture was then heated to reflux under an atmosphere of nitrogen gas for 4 h. After this time a yellow suspension was observed which was isolated by filtration, washed with ethanol followed by ether to yield a yellow powder (156.2 mg, 0.38 mmol, 78%). ESI–MS: [C_20_H_23_N_6_O_2_S]^+^
*m/z* 411.16 (experimental), 411.16 (calculated), HPLC: Rt = 12.45 min, ^1^H NMR: (500 MHz; d_6_-DMSO): δ 10.27 (s, 1H, CH_3_NHC=SN*H*), 10.17 (s, 1H, Pyr-N*H*), 8.72 (dd, *J* = 2.4, 0.7 Hz, ArH), 8.33 (d, *J* = 4.6 Hz, 1H, CH_3_N*H*), 8.14 (dd, *J* = 8.8, 2.4 Hz, 1H, ArH), 7.50 (d, *J* = 8.9 Hz, ArH), 7.36 (dd, *J* = 8.8, 0.6 Hz, ArH), 7.26 (d, *J* = 0.8 Hz, ArH), 7.13 (d, *J* = 2.6 Hz, ArH), 6.87 (dd, *J* = 8.9, 2.6 Hz, ArH), 3.80 (s, 3H, CH_3_), 3.04 (d, *J* = 4.6 Hz, 3H, NHC*H_3_*), and 2.25 (m, 6H, 2 × CH_3_). ^13^C NMR (126 MHz; DMSO-d_6_): δ 178.4, 156.9, 155.8, 154.4, 148.9, 148.5, 145.5, 144.4, 134.3, 129.6, 118.4, 112.5, 111.4, 107.1, 103.3, 100.7, 55.57, 55.55, 31.2, 11.5, and 11.1.

#### 4.3.3. [Copper(II)(Diacetyl-2-(2-hydrazino-pyridinyl-5-(5′-methoxybenzofuran))-(4-methyl-3-thiosemicarbazone))] (CuL^5^)

A suspension of H_2_L^5^ (60.5 mg, 0.15 mmol) and copper acetate monohydrate (37.0 mg, 0.18 mmol) in ethanol (10 mL) was heated to reflux under an atmosphere of nitrogen gas for 2 h, and then stirred at ambient temperature for 16 h. The deep purple mixture was then filtered and a purple solid washed with ethanol followed by diethyl ether. The solid was then air dried to yield a dark purple powder. (59.4 mg, 0.13 mmol 87% yield). ESI–MS: [C_20_H_21_CuN_6_O_2_S]^+^
*m/z* 472.08 (experimental), 472.07 (calculated). HPLC: Rt = 13.06 min.

### 4.4. Cell Stimulation and/or Treatment

#### 4.4.1. Preparation of Metal Complexes, Copper-Free Ligand and Inflammatory Mediators and Inhibitors

Reagents were prepared fresh prior to cell stimulation and treatment procedures. Cytokines, including TNFα (Life Technologies, Carlsbad, CA, USA) and IFNγ (Life Technologies, Carlsbad, CA, USA) were diluted to concentrations of 10 and 15 ng/mL together in the appropriate cell growth media. Copper(II) complexes (Figure 1) and minocycline hydrochloride (Sigma-Aldrich, Burlington, MA, USA) were resuspended to 10 mM in ultrapure H_2_O or dimethyl sulfoxide (DMSO) Hybri-Max™ (Sigma-Aldrich, Burlington, MA, USA), as appropriate. Copper- complexes and minocycline hydrochloride were then further diluted to the final concentrations in appropriate cell growth media, alone or in combination with TNFα and IFNγ.

#### 4.4.2. Stimulation and Treatment of Cell Cultures

BE(2)-M17 or microglial cultures were treated with varying concentrations (0.5–20 μM) of copper(II) complexes for 24 h. Cultured cells were also treated with minocycline hydrochloride (20 μM) as an anti-inflammatory control. Treatments of microglial cultures were also carried out concurrently with TNFα and IFNγ stimulation as described in [34]. All cell stimulations and/or treatments were performed in fresh cell growth media. After treatment, conditioned media were collected for lactate dehydrogenase (LDH) cytotoxic assay and ELISA while cells were used for 3-(4,5-dimethylthiazol-2-yl)-2,5-diphenyltetrazolium bromide (MTT) reduction assay, qRT-PCR, or collected for inductively coupled plasma mass spectrometry (ICP–MS), as appropriate.

### 4.5. 3-(4,5-Dimethylthiazol-2-yl)-2,5-Diphenyltetrazolium Bromide (MTT) Reduction Assay

Cell viability was determined using the 3-(4,5-dimethylthiazol-2-yl)-2,5-diphenyltetrazolium bromide (MTT) reduction assay, based on the ability of viable cells to reduce MTT to the coloured compound, formazan. Conditioned media in sample wells on 24 well culture plates was replaced with 300 µL of fresh media supplemented with MTT (480 μM; Sigma-Aldrich, Burlington, MA, USA) and incubated at 37 °C until sufficient MTT conversion to formazan was achieved. At 30 min prior to performing the assay, Triton X-100 (TX, 1% (*v*/*v*); Merck Millipore, Burlington, MA, USA) was added to several wells of BE(2)-M17 cells as a positive control for cell death (i.e., no MTT conversion). Following 30 min of colour development, plates were centrifuged (1000× *g*, 3 min, RT) and media were aspirated. A volume of 300 µL of DMSO was then added to individual wells to resuspend the formazan crystals. A volume of 100 μL of each sample in duplicates was transferred onto a 96-well microplate (Greiner Bio-One, Frickenhausen, Germany) and the absorbance at 585 nm was determined with an EnSpire^®^ Multimode Plate Reader (Perkin Elmer, Waltham, MA, USA). Following blank subtraction, the percentage cell viability relative to non-treated (media-only) controls was calculated. Data are presented as the percentage MTT reduction compared to media-only control.

### 4.6. LDH Cytotoxic Assay

Cell viability was measured following cell treatment using the Cytotoxicity Detection Kit (Roche, Basel, Switzerland) according to the protocol provided by the manufacturer. At 30 min prior to performing the assay, 1% (*v*/*v*) TX was added to several wells BE(2)-M17 cells as a positive control for LDH release. A volume of 100 μL of each media sample or control was mixed with 100 μL of LDH reaction mixture (prepared by mixing reconstituted catalyst and dye solution, provided in the kit, in a 1:46 ratio) in individual wells of a 96-well microplate. Absorbance at 490 nm was determined following colour development for 30 min in the dark at RT using an EnSpire^®^ Multimode Plate Reader. Following blank subtraction, the percentage LDH released was calculated relative to TX treated positive controls. Data are presented as the percentage of LDH released relative to TX-treated control.

### 4.7. Inductively Coupled Plasma Mass Spectrometry (ICP–MS)

BE(2)-M17 cells were washed and collected in Tris buffered saline (TBS; 150 mM NaCl and 50 mM Tris; pH 7.4) and centrifuged at 326× *g* for 10 min at 4 °C. Cell pellets were then resuspended in 110 μL of TBS. A volume of 100 μL of cell-containing TBS was transferred into a new centrifuge tube and further centrifuged for 10 min at 326× *g*. Following the removal of the supernatant, cell pellets were stored at −20 °C prior to ICP–MS. To measure brain tissue copper content in 5xFAD and non-transgenic mice, animals were euthanized by CO_2_ at experimental end-point (1 week gavage). The brain stem and olfactory bulbs were removed, brains were dissected into hemispheres and snap frozen using liquid nitrogen and stored at −80 °C until use for copper content analysis by ICP–MS. ICP–MS analyses were performed at The Florey Institute of Neuroscience and Mental Health as previously described in [34,108]. For standardization of the ICP–MS results to protein concentrations, 10 μL of cell-containing TBS was added to 20 μL of PhosphoSafe™ Extraction Reagent (Merck Millipore, Burlington, MA, USA) (with added 5% (*v*/*v*) phenylmethyl sulfonyl fluoride (PMSF) (Thermo Scientific, Waltham, MA, USA) and 1% (*v*/*v*) DNase 1 (Roche, Basel, Switzerland) and stored at −20 °C until used for bicinchoninic colorimetric assay (BCA) to determine the protein concentration. BCA was carried out using the BCA Protein Assay Kit (Thermo Scientific, Waltham, MA, USA) as per manufacturer’s protocol. In brief, bovine serum albumin (BSA) standards (0–2 mg/mL) were prepared by serial dilution. In a 96-well microplate, 10 μL of standards or samples, in duplicate, was mixed with 190 μL of BCA Protein Assay Reagent mixture (prepared by mixing BCA Protein Assay Reagents A and B in a 50:1 ratio). Absorbance at 560 nm was determined following 15 min colour development in the dark at RT using an EnSpire^®^ Multimode Plate Reader. Standard curves were constructed to determine sample protein concentrations. By standardisation of raw data obtained from ICP–MS to protein concentrations as determined using BCA assay, copper concentrations in cells were expressed as μg copper/g protein. For normalisation of results obtained for brain tissues ICP–MS, copper content determined for each sample in μg was standardised to wet tissue weight in g, measured prior to snap-freezing.

### 4.8. ELISA

Media collected from microglial cell cultures were centrifuged at 6000× *g* for 5 min at 4 °C. Supernatants were stored at −80 °C until used for ELISA. MCP-1 ELISA was carried out using Mouse CCL2/JE/MCP-1 DuoSet as per the protocol described by R&D Systems. Absorbance at 450 nm was determined using an EnSpire^®^ Multimode Plate Reader. Standard curves were constructed, and sample cytokine concentrations were calculated from standard curves. Results obtained were expressed as percentage secretion relative to the TNFα- and IFNγ-stimulated positive controls for each cell type.

### 4.9. qRT-PCR

RNA extraction from microglia and cDNA synthesis was carried out using the TaqMan^®^ Gene Expression Cells-to-CT™ Kit (Thermo Scientific, Waltham, MA, USA) as per the manufacturer’s protocol. Individual wells of microglia were washed in 100 μL of ice-cold PBS. A volume of 50 μL of lysis solution (containing DNase 1 diluted 1:100) was added to each well and incubated for 5 min at RT. A volume of 5 μL of stop solution was then added into each well and incubated for 2 min at RT. Samples were then used immediately for cDNA synthesis or stored at −80 °C. For cDNA synthesis, 50 μL reactions in 0.2 mL tubes contained 50% (*v*/*v*) 2xRT buffer, 5% (*v*/*v*) 2xRT enzyme mix, up to 45% (*v*/*v*) RNA sample and nuclease-free water, as appropriate. Thermal cycling conditions for reverse transcription were: 60 min at 37 °C and 5 min at 95 °C. cDNA samples were then used immediately for qRT-PCR or stored at −20 °C. Each qRT-PCR reaction mix was made up to a final volume of 10 µL with 2 µL cDNA, 5 µL TaqMan^®^ Advanced Master Mix (Life Technologies, Carlsbad, CA, USA), 0.5 µL appropriate gene assay mix (including *Hprt* (Assay ID: Mm01545399_m1), *Mt1* (Assay ID: Mm00496660_g1), *Tnf* (Assay ID: Mm00443258_m1), *Cd33* (Assay ID: Mm00491152_m1) and *Trem2* (Assay ID: Mm04209424_g1)) (Life Technologies, Carlsbad, CA, USA) and 2.5 µL RNase-free water in a 384-well PCR plate (Applied Biosystems, Waltham, MA, USA). Plates were then sealed, centrifuged (10,000× *g*, 1 min at RT), and loaded into a QuantStudio™ 6 Flex Real-Time PCR System (Applied Biosystems, Waltham, MA, USA) to perform qRT-PCR under the following conditions: 50 °C for 2 min, 95 °C for 10 min and 40 cycles of 15 sec at 95 °C and 1 min at 60 °C. Cycle threshold (Ct), the minimum cycle number required to produce an exponential increase in fluorescence, was determined for each reaction using the second derivative method. Results obtained for *Mt1*, *Tnf*, *Trem2* and *Cd33* were normalised to gene expression level of *Hprt*. Results were presented as fold induction relative to media-only controls.

### 4.10. Phagocytosis Assay

Primary microglia were seeded at 12,000 cells/well in a 96-well plate. To examine the effect of cell stimulation and/or treatment on microglial phagocytic properties, pHrodo^®^ Red *E. coli* BioParticles^®^ Conjugate (Thermo Scientific, Waltham, MA, USA) uptake by microglia was carried out with modification from manufacturer’s instruction. A stock solution was prepared by resuspending pHrodo^®^ Red *E. coli* BioParticles^®^ Conjugate in microglia growth media (2 mg/mL). Cells were added with pHrodo^®^ Red *E. coli* BioParticles^®^ Conjugate (66.7 μg/mL final concentration) and imaged using an IncuCyte ZOOM system (Essen Bioscience, Ann Arbor, MI, USA) incubated at 37 °C and 5% CO_2_. Four images per well were taken every 2 h for 24 h using a ×10 objective lens with phase-contrast and red fluorescence options. Red channel acquisition time was 400 ms. Quantification was performed using basic analysis option provided in IncuCyte ZOOM software (2016A). Red channel background was filtered out by applying a Top-Hat method, with a correction radius of 20 µm, and a threshold of 2 red corrected units. Edge split was applied with closely placed objects to enable a more accurate quantification of red fluorescence. We used red object count per image as the outcome measure. Same settings were applied to all images. Cytochalasin D (Sigma-Aldrich, Burlington, MA, USA), an inhibitor of actin polymerization was used at 10 µM to reduce pHrodo-labeled uptake.

### 4.11. Oral Treatment of 5xFAD with CuL^5^

Prior to treatment, male 5xFAD and non-transgenic mice were randomised into treatment groups as required for the experimental set-up: Sham and CuL^5^. Treatments were allocated evenly across litters and genotypes in each independent trial. Sham treated mice were subjected to oral gavage with standard suspension vehicle (SSV; 0.9% (*w/v*) NaCl (Merck Millipore, Burlington, MA, USA), 0.5% (*w/v*) Na-carboxymethylcellulose (Sigma-Aldrich, Burlington, MA, USA), 0.5% (*v*/*v*) benzyl alcohol (Sigma-Aldrich, Burlington, MA, USA), 0.4% (*v*/*v*) Tween-80 (Sigma-Aldrich, Burlington, MA, USA)). For treatment with CuL^5^, mice were orally gavaged with CuL^5^ resuspended in SSV, homogenised by sonication prior to treatment. CuL^5^ were administered to each animal at a dose of 30 mg/kg bodyweight. Treatments were administered once daily from 10 weeks of age, 7 days/week, for 1 or 8 weeks as indicated. At experimental endpoint, mice were euthanised by CO_2_ for blood collection or transcardial perfusion with ice-cold PBS for the collection of whole brain samples. Brain tissues were collected and measured for copper content by ICP–MS, used for ex vivo microglia isolation or histological processing.

### 4.12. Biochemical Analysis of Serum

At the end of a week of dose-escalation trial by gavage, mice were euthanized by CO_2_ and blood samples from individual mice were collected by cardiac puncture. EDTA (Sigma-Aldrich, Burlington, MA, USA; 0.05 mM final concentration) was immediately added to individual samples as an anti-coagulant. Samples were then centrifuged at 1000× *g* for 15 min at 4 °C. Blood plasma samples in the upper phase were than collected into a fresh Eppendorf tube. Plasma samples were stored at −20 °C and sent to ASAP Laboratory (Mulgrave, VIC, Australia) for biochemical profile analysis.

### 4.13. Acute Microglia Isolation and Analysis

2 h prior to euthanasia, mice were injected intraperitoneally with Methoxy–X04, a red Congo derivative that labels the Aβ deposits in the brain (2 mg/mL in 1:1 ratio of DMSO to 0.9% (*w/v*) NaCl, pH 12) at 5 mg/kg. Mice were deeply anesthetized with tribromoethanol (Sigma-Aldrich, Burlington, MA, USA) and transcardially perfused with ice-cold heparinised saline prior to brain extraction. Microglia isolation was performed as described in [59]. One hemisphere of a brain, excluding brain stem and olfactory bulbs, were dissected into cerebellum and non-cerebellum regions for microglia isolation. Single cell suspensions were prepared from brain tissues by mechanical dissociation using mesh of decreasing sizes from 250 μm to 70 μm and enriched for microglia by density gradient separation [59]. The cell pellet was resuspended in 70% (*v*/*v*) isotonic Percoll (1× PBS + 90% (*v*/*v*) Percoll, Burlington, MA, USA), overlayed with 37% (*v*/*v*) isotonic Percoll and centrifuged with slow acceleration and no brake at 2000× *g* for 20 min at 4 °C. The microglia-enriched cell population isolated from the 37–70% interphase was diluted 1:5 in ice-cold PBS and recovered by cold centrifugation at maximum speed for 1 min in microcentrifuge tubes. The cell pellet was then stained with antibodies to microglial cell surface markers (CD11b-BV650; 1:200; Biolegend #141723, CD45-BV786; 1:200; BD Biosciences #564225, CX3CR1-FITC; 1:100; Biolegend #149019) for isolation using the FACSAria™ III cell sorter. Microglia were defined as live/propidium iodide (PI)^−^ (Sigma-Aldrich, St. Louis, MO, USA, #P4864), CD11b^+^, CD45^mid^, CX3CR1^+^ single cells. The XO4^+^ population gate was set using Methoxy-XO4-injected wild-type animals. For each sample from individual animal, the distribution of microglia populations was expressed as percentage X04^+^ microglia, relative to total microglial count.

### 4.14. Immunohistology

The other hemisphere was fixed in 4% (*w/v*) PFA (Santa Cruz Biotechnology, Dallas, TX, USA) overnight, followed by 48 h immersion in 30% (*w/v*) sucrose solution then frozen using liquid N_2_. Samples were stored at −80 °C prior to sectioning. Frozen brain hemispheres were cryostat-sectioned into 60 μm thick sagittal sections, collected starting at the hippocampus onto Superfrost™ Plus slides (Thermo Scientific, Waltham, MA, USA) in series for histological staining. A total of 4 tissue sections per animal, ~600 μm apart, representing the hippocampus progression were blocked for 1.5 h at RT in PBST (containing 2% (*w/v*) BSA and 0.5% (*v*/*v*) TX). Sections were then stained overnight at RT for microglia with the rabbit anti-Iba1 (1:500; WAKO #019-19741) primary antibody. Following 3 × 5 min washes in PBST, the AlexaFluor 488 goat anti-rabbit IgG (H+L) secondary antibody (1:500; #A11008; Life Technologies, Carlsbad, CA, USA) was added for 2 h incubation at RT. After 3 × 5 min washes in PBST, sections were mounted in Prolong Diamond Antifade Mountant. Sections were imaged by a blinded investigator with an Olympus IX71 inverted microscope using the 10× objective.

### 4.15. Imaging and Image Analyses

Amyloid plaque and microglia staining were quantified from the hippocampal region of the 3rd tissue section on each slide from individual animal. Me-X04 and Iba1 fluorescent images in tif format were batch processed using Fiji ImageJ (Wayne Rasband; National Institutes of Health, Bethesda, MA, USA) version 1.50a. Percentage Me-X04 and Iba1 immunoreactivity was quantified as the percentage of positive pixels for pixel intensity ranging between 80 and 255, and 110 and 255, respectively. Using the thresholding algorithms, images were batched converted to binary masks. Plaque counts were quantified using the 3D Objects Counter algorithm in Fiji. Data are presented as the mean of *n* ≥ 5 individual animals per treatment group.

### 4.16. Behavioral Battery

#### 4.16.1. Nest-Building Test

Nest building was assessed by supplying the home cage of each mouse with a piece of paper towel (Serla, Finland, size of 22 × 22 cm). Environmental enrichments, including igloo and nesting material, sawdust bedding, food and water were normally available. The next morning (approximately 21 h later), the cages were inspected for nest construction. Nests were photographed and documented. Paper towel constructions were scored between 0–3 points as follows: 0 points = an intact nest without or with a few bites or tears on paper, 1 point = a flat nest with few bites or tears, 2 points = moderately torn up and bitten or torn nest and 3 points = a torn up nest in the form of a crater or as a part of other nesting material as described previously [109] (Appendix A).

#### 4.16.2. igloo Test

This test is a version of rewarded delayed alternation paradigm to measure spatial working memory of the mouse in its home cage. The test utilizes common plastic enrichment shelters for mice, ‘igloos’. First, the home cage lid, nesting material and the normal igloo were removed. Two plastic igloos (diameter 10 cm, Bio-Serv, Flemington, NJ, USA) with only one entrance (two other entrances were sealed by opaque tape) and an opaque partition wall were placed in the home cage.

A rice cereal was used as a food reward that was alternately inserted between the two igloos so that the igloo that was not last visited was rewarded. In other words, after the mouse had entered a rewarded igloo, a new reward was inserted to the other igloo. In the beginning of the practise period, the rice cereal was placed near the entrance and gradually inserted in the center of the igloo. After entering one of the two igloos, mice were gently pushed behind the partition wall. Between each choice, there was 20 s long delay when the mouse was behind the partition wall waiting and eating the rice cereal if it was found and successfully carried. The partition wall and test igloos were cleaned with 70% ethanol after each mouse.

Mice were trained once a day for 10 trials and for five consecutive days. After two trial-free days, the first test period was performed according to the same principles. The first test period was conducted in males at the age of 8.5 months and around nine months of age in females. From six to nine weeks after the first period, the second test period was conducted (in males around the age of 11 months and in females at the age of 10.5 months) in a similar fashion as the first test period. The correct and incorrect choices and perseverative errors (three or more consecutive incorrect choices) were recorded on each test day.

The mice were food restricted diet to motivate them to search for the rice cereal. The day before the practice period or test periods, normal food was removed and food pellet the size of 1.0–1.3 g was inserted in the home cage. After each practice and test day, the mouse was given one food pellet and 1–1.5 heaped teaspoon of rice cereals. During the trial-free intervals and after the test periods, the mice received food ad libitum. No weight reduction above 10% was allowed.

#### 4.16.3. Morris Water Maze Task

A circular pool (diameter 120 cm, height 22 cm, painted white) was filled with tap water (20–25 °C). Four spotlights, a black screen and other different shaped objects were provided as distal spatial clues for mice (Appendix A). A square, clear platform (size of 10 × 10 cm) was placed in the in the middle of South-East target quadrant submerged about 1 cm below the water surface. A video camera was attached on the ceiling in the center of the pool. Swimming speed and dwell time in different pool locations were analyzed with EthoVision XT 7.0 (Noldus Information Technologies, Wageningen, The Netherlands) video tracking software.

About 3 days before the actual acquisition trials, mice were familiarized with test conditions guided by a high-walled alley (length 70 cm, width 13 cm, painted black), inside the pool (Appendix A). Each trial lasted until the mouse had found the hidden platform and stayed on it for 10 s. This pre-training was repeated alternatingly from both ends of the alley in total four times a day on two subsequent days.

Acquisition trials (five trials a day for five days) were assessed by placing a mouse in the water facing the pool wall from one of the four start positions (N, E, S, W) in a predefined varying order each day. If the mouse did not find the platform within 60 s, it was gently lifted to the platform and placed on it for 10 s. Between the trials, the mouse was dried and placed on towels in a heated cage for 5–10 min breaks. After the last trial of the day, the mouse was returned to its home cage.

On the fifth day (~24 h after the first acquisition trial of the previous test day), the platform was removed, and a probe trial of 60 s was started from the opposite position from where the platform had located. The time spent in the platform zone and each quadrant, the average distance from the platform zone, rotation, speed, and the number of times the mouse crossed the location where the platform had been located were recorded. After the probe trial, the platform was placed back in its position and mice were given three acquisition trials from the other three starting positions. Finally, the second probe trial was conducted in a similar way as the first probe and after this, the mouse was returned to its home cage.

### 4.17. Statistical Analysis

Statistical analysis was performed using GraphPad Prism version 9.2.0. Data were analyzed using an unpaired two-tailed *t*-test when comparison between two groups was being investigated or one- or two-way ANOVA followed by post hoc tests when comparisons between three or more groups were analysed, with *p* value of less than <0.05 considered statistically significant. MWM data were analysed using mixed-model ANOVA for repeated measures. Results are shown as the mean±SEM unless specified differently in the figure legends. The number of independent (n) replicates used for each experiment are specified in figure legends.

## 5. Conclusions

Here, we report on the identification of a new copper(II) complex (CuL^5^) with the potential to modulate neuroinflammatory changes in AD. There are obvious limitations in this report that need to be further addressed in future studies. These include the need to understand why CuL^5^ only improved behavioural function in one of several cognitive behaviour tests, why there was a paradoxical increase in amyloid accumulation in vivo despite improved phagocytosis in vitro, and what the relationship is between localized copper homeostasis and CuL^5^ effects. From the experiments described here, the speculative screening of a very small series of small simple copper(II) complexes identified CuL^5^ as having the ability to modulate cytokine-induced inflammation and phagocytosis, as well as having direct impact on two important microglial LOAD risk genes, *Cd33* and *Trem2*. These results provide further support that membrane-permeant copper(II) complexes can potentially offer improvements to neuroinflammation in neurodegenerative diseases such as AD. Although the therapeutic benefit in this study was restricted to some effects on spatial memory, the findings support the potential further development of copper(II) complexes aimed at targeting the neuroinflammatory changes in AD.

## Figures and Tables

**Figure 1 ijms-23-10722-f001:**
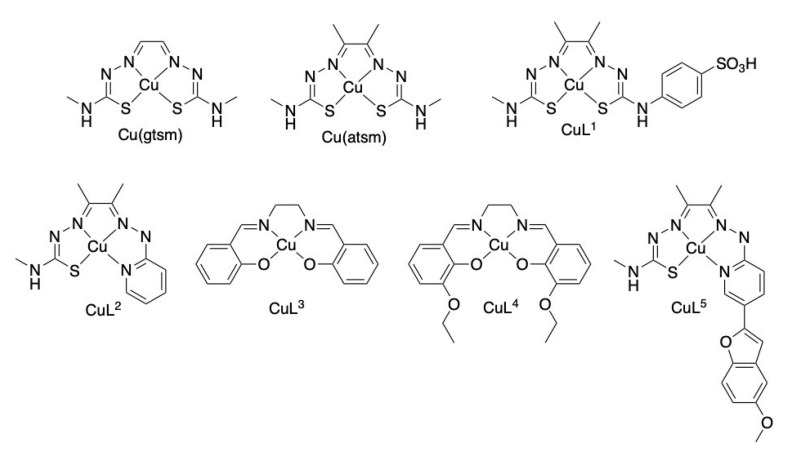
Chemical structures of tested copper(II) compounds Cu^II^(gtsm), Cu^II^(atsm), CuL^1−5^.

**Figure 2 ijms-23-10722-f002:**
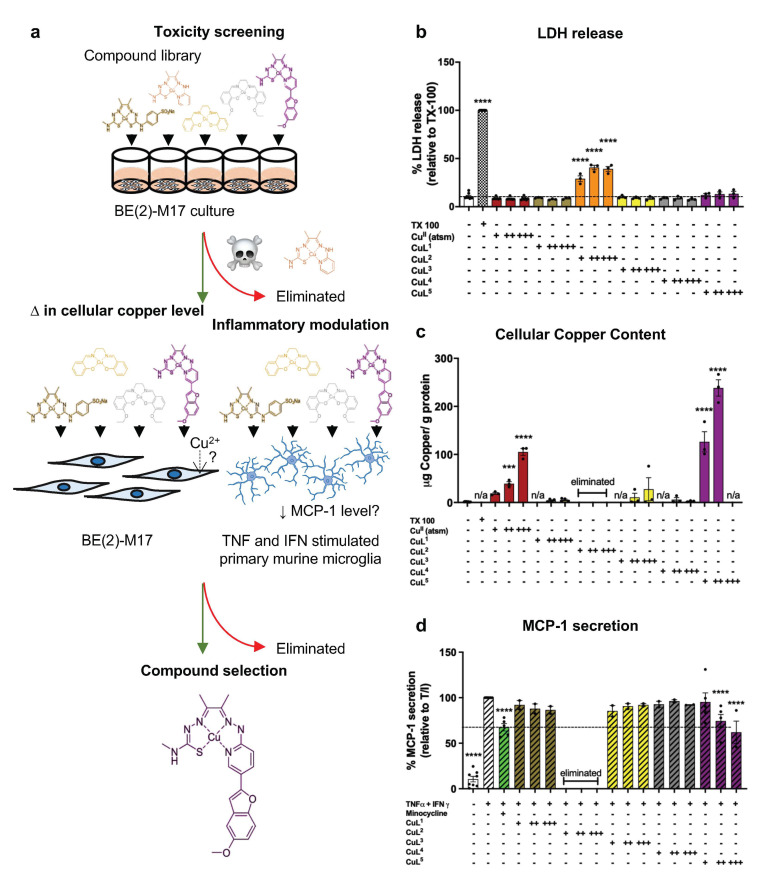
CuL^5^ treatment exhibited inflammation-modulating effect. (**a**) Workflow of the screening pipeline to identify leading compound(s). (**b**,**c**) M17 cultures were incubated for 24 h with new-generation metal compounds (2–10 μM). (**b**) Cell viability and (**c**) cell–copper association were determined by LDH assay and ICP–MS, respectively. (**d**) Primary murine microglial cultures were stimulated with TNFα (10 ng/mL) and IFNγ (15 ng/mL) and concurrently treated with 20 μM minocycline or varying concentrations of CuL^1^, CuL^2,^ CuL^3^, CuL^4^, or CuL^5^ for 24 h. MCP-1 secretion by microglia was measured by ELISA. Data are expressed as (**b**) percentage LDH released relative to Triton X-100 (TX 100)-treated positive control, (**c**) μg copper per g cellular protein, and (**d**) percentage of MCP-1 released relative to TNFα- and IFNγ-stimulated positive control and presented as the mean ± S.E.M. Dotted line represents (**b**) average % LDH release of the media-only negative control and (**d**) average % MCP-1 released when TNFα- and IFNγ-stimulated microglia were treated with minocycline. Determined by one-way ANOVA, compared to (**b**,**c**) media-only or (**d**) TNFα- and IFNγ-stimulated controls ***/**** = *p* ≤ 0.001/0.0001. Numbers on *x*-axes represent μM compound. n/a = concentration not assessed.

**Figure 3 ijms-23-10722-f003:**
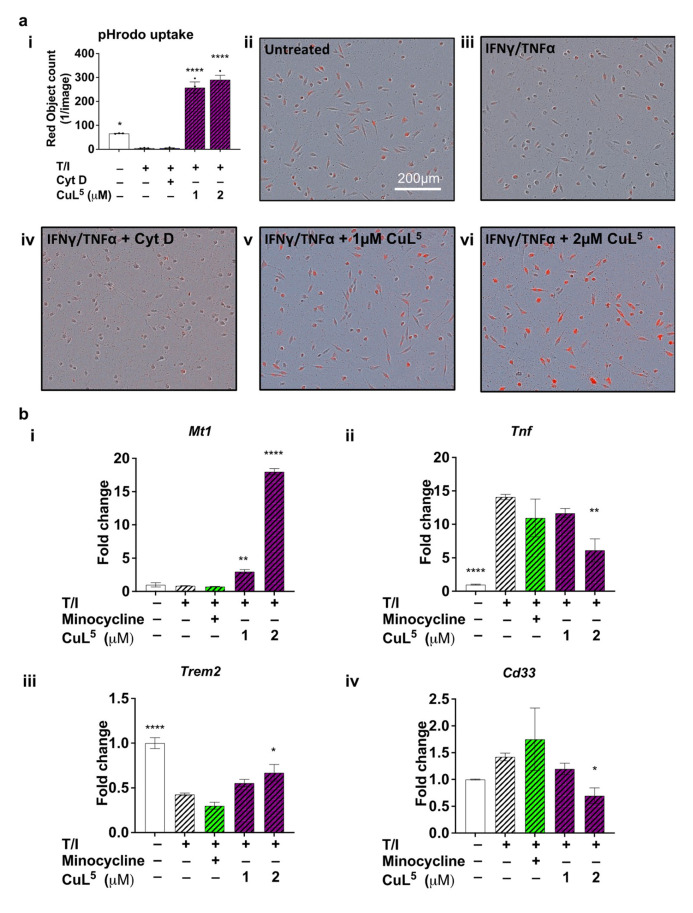
CuL^5^ treatment modulated microglial phagocytic activity. Primary murine microglial cultures were stimulated with TNFα (10 ng/mL) and IFNγ (15 ng/mL) and concurrently treated with 10 µM cytochalasin D, 20 µM minocycline, 1 µM CuL^5^, or 2 µM CuL^5^ for up to 24 h. (**a**) 1 h post-stimulation and treatment, media were removed and replaced with 100 µg/mL pHrodo red *E.coli* bioparticles and fluorescence emission was measured using the IncuCyte imaging platform every 2 h. (**aii**–**avi**) Representative images from one experiment showing phase contrast and red channel signal with respective conditions. An actin inhibitor, cytochalasin D (10 µM) was incubated 1 h prior to the addition of *E.coli* bioparticles to validate pHrodo-labeled uptake. Scale bar = 200 µm. (**b**) The 24 h post-stimulation and treatment, RNA was extracted from cells for cDNA synthesis. qRT-PCR was used to determine mRNA expression levels of (**bi**) *Tnf*, (**bii**) *Mt1*, (**biii**) *Trem2* and (**biv**) *Cd33*. Results obtained were normalised to mRNA expression level of *Hprt*. (**ai**) Data are expressed as the average red object count (1/image) of pHrodo-labeled microglial cells in each condition. Three images per well from three technical replicates were analysed. (**b**) Representative data set is expressed as fold induction relative to media-only control. Data are presented as the mean ± S.E.M. Determined by one-way ANOVA, compared to (**bi**) media-only or (**ai**,**bii**–**biv**) TNFα- and IFNγ-stimulated controls, */**/**** = *p* ≤ 0.05/0.01/0.0001.

**Figure 4 ijms-23-10722-f004:**
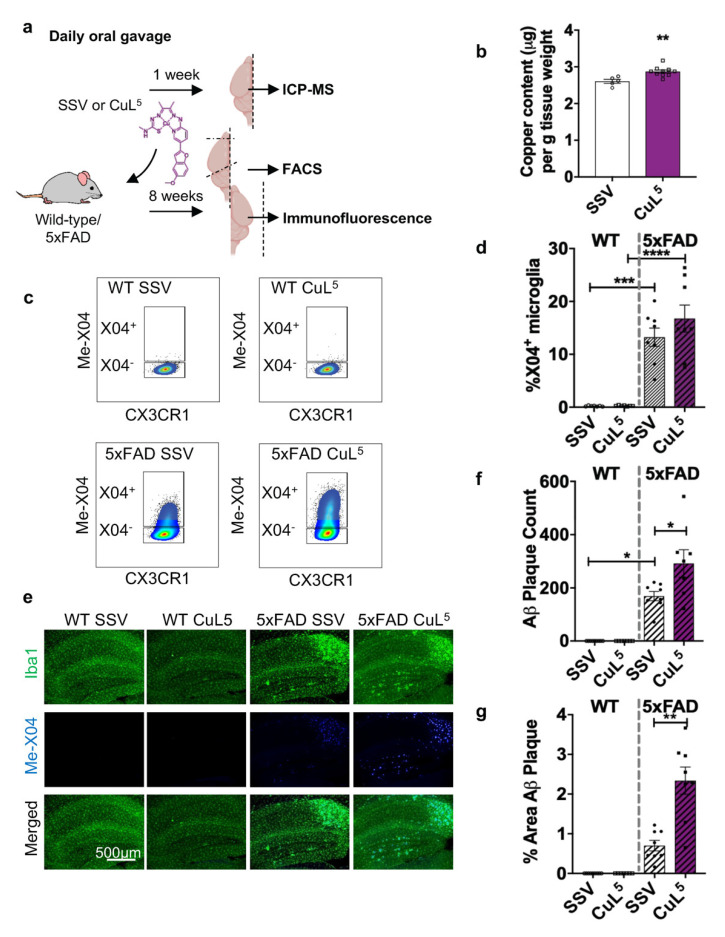
CuL^5^ treatment increased microglial phagocytosis and A*β* load in 5xFAD mice. (**a**) Illustration of experimental workflow, created with BioRender.com. SSV or CuL^5^ (30 mg/kg) were delivered to non-transgenic (WT) or 5xFAD mice by oral gavage, once daily, for 1 or 8 weeks. (**b**) Brain samples collected from a mixed cohort of WT (*n* = 10) and 5xFAD (*n* = 5) animals treated for 1 week were analysed for copper content by ICP–MS (*n* = 5–10 per treatment group). Individual mice treated for 8 weeks were injected with Methoxy-X04 prior to the collection of brain samples for (**c**,**d**) ex vivo isolation of microglia and analysed for the proportion of X04^+^ and X04^−^ microglia by FACS (*n* = 7–8 per treatment group) and (**e**–**g**) the analysis of Aβ plaque load and microgliosis by immunohistochemistry (*n* = 7–8 per treatment group). (**b**) Brain copper content measured by ICP–MS is expressed as copper content (μg) per g brain tissue. (**c**) Representative FACS plot for microglia sorted from WT and 5xFAD mice, from each treatment group. (**d**) Data obtained are expressed as percentage of X04^+^ microglia population in total microglia sorted. (**e**) Representative immunofluorescence image of the hippocampus of WT and 5xFAD mice, from each treatment group, injected with Methoxy-X04 and stained for Iba1 (AlexaFluor 488). The (**f**) number of plaques and (**g**) percentage area of plaques in the hippocampal region were determined using ImageJ. (**b**,**d**,**f**,**g**) Data for each treatment group are presented as the mean ± S.E.M per treatment group. Determined by one-way ANOVA, */**/***/**** = *p* ≤ 0.05/0.01/0.001/0.0001.

**Figure 5 ijms-23-10722-f005:**
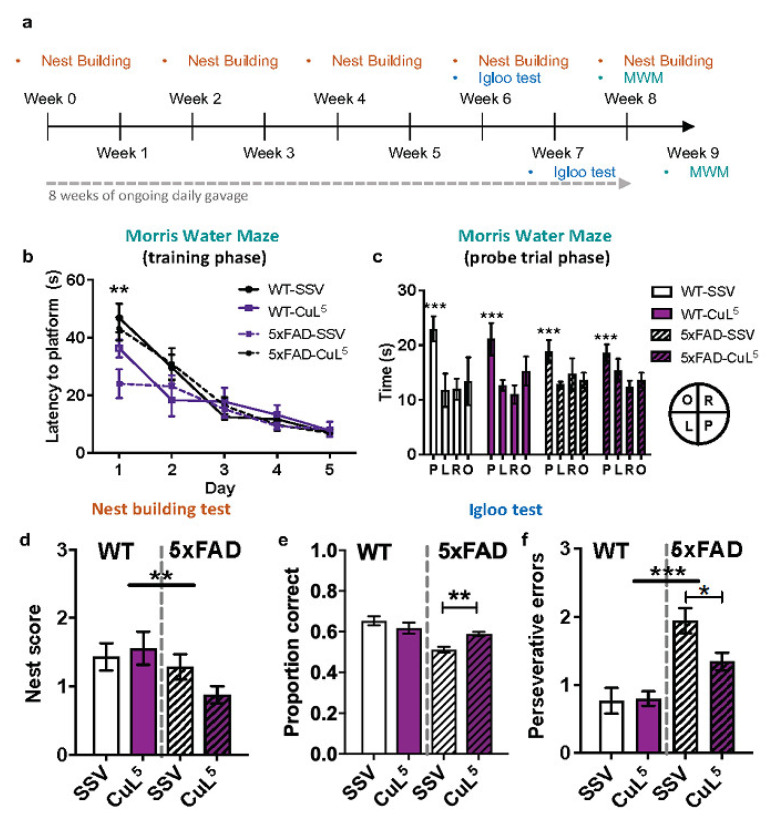
Effect of CuL^5^ treatments on 5xFAD cognitive functions. (**a**) Timeline for behavioural testing of CuL^5^-treated mice. SSV or CuL^5^ (30 mg/kg) were delivered to non-transgenic (WT) or 5xFAD mice by oral gavage, once daily, for 8 weeks while also being to a battery of behavioural tests to assess the effect of treatment (*n* = 6–7 per treatment group). (**b**) Escape latency in the Morris water maze during the training phase. (**c**) Time spent in the former platform quadrant (P) and quadrats opposite (O), to the left (L), and to the right (R) of the platform during the probe trial. (**d**) Nest quality scores. (**e**) Choice accuracy in igloo test. (**f**) Perseverative errors in igloo test. Data for each treatment group are presented as the mean ± S.E.M per treatment group. Morris water maze data and choice accuracy in the igloo test were analysed with two-way ANOVA for repeated measure with Bonferroni pairwise comparisons. Perseverative errors in the igloo test and nest building scores estimated by Poisson regression as incidence rate ratios (IRRs), and by logistic regression as odds ratios (ORs), respectively. */**/*** = *p* ≤ 0.05/0.01/0.001.

## Data Availability

All data are presented in the main text or the Appendix A or can be made available by contacting the corresponding author.

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
