# Peer review of "Novel Anti-Neuroinflammatory Properties of a Thiosemicarbazone–Pyridylhydrazone Copper(II) Complex"

_ijms, 2022, doi:10.3390/ijms231810722_

Round 1

Reviewer 1 Report

Both (Cu(gtsm) and Cu(atsm)) have been extensively examined for broad medicinal and therapeutic actions in preclinical models of neurodegeneration. In this manuscript, Choo et al have demonstrated that copper-complex of thiosemicarba-zone-pyridylhydrazone, particularly CuL5, 

can induce anti-neuroinflammatory effects in vitro and provide selective benefit in vivo. They acknowledge that they have many questions left unanswered from this study, it perhaps accurately reflects how our understanding of AD and other brain disorders is still lacking, and that all efforts towards understanding how copper medicines affect AD are worthwhile endeavours. Overall, the data are solid (in fact, supported by large amounts of data) and convincing. This reviewer would recommend this article for publication prior to some minor revision.

1. It may be worth briefly commenting on how the authors' data and results line up with some of the new AD pathogenesis models (although controversial). For example, how their lead complex has anti-neuroinflammatory effects in vitro but only selective benefits in vivo could give insight as to whether neuroinflammation is a cause or a consequence of neurodegeneration in AD.

2. A paragraph on 'conclusion and perspective' shall be added after the 'discussion' to further attract more readers of the journal.

Reviewer 2 Report

The past two decades of research into the pathogenesis of Alzheimer disease (AD) have been driven largely by the amyloid hypothesis; the neuroinflammation that is associated with AD has been assumed to be merely a response to pathophysiological events. However, new data from preclinical and clinical studies have established that immune system-mediated actions in fact contribute to and drive AD pathogenesis. These insights have suggested both novel and well-defined potential therapeutic targets for AD, including microglia and several cytokines. Therefore, the presented manuscript includes an interesting study which determine the anti-neuroinflammatory action of several low molecular weight, charge neutral and lipophilic copper-complexes. This study is timely and scientifically sound and properly written, following all the guidelines for publications of scientific articles and it is within the scope of the journal. The abstract of the manuscript is written in great detail and fully covers all parts of the manuscript. The figures additionally facilitate the full evaluation of the content contained in the manuscript and significantly increase its value. The introduction provides an interesting admission to the topic. The methods are described in a detailed and clear manner. The discussion contains the most important information necessary to draw conclusions from the conducted research. I can only suggest that the last paragraph of introduction should be placed as a summary of the work, and the place where the authors added it should only be the purpose of the work. Forthermore, the exact conditions of detention of the animals and the size of the groups should be added.

In conclusion I wish all participating Authors a lot of success and energy in further creative research work in such a socially important area as the fight against Alzheimer disease.

Reviewer 3 Report

1The authors have investigated the immuno-modulation potency of five copper-complexes in microglia. Based on invitro assays of cytotoxicity and modulation of microglial phagocytic activity, one complex was further CUL5 was further evaluated in vivo. The authors have found increased accumulation of copper ions in brain tissue at 8wks post gavage administration of the complex. Furthermore studies with 5X FAD mice show enhanced phagocytic migroglial population. Interestingly, the compound appears to enhance hippocampal plaque load. A battery of behavior tests found an improvement in short term memory functions. The complex thus have therapeutic potential but needs extensive study regarding its mechanism of function.

Comments:

1. It is unclear why only one of the tested copper complex was modified to make it water soluble. Were the other 4 complexes water soluble? It will be difficult to make a proper comparison of their efficacies if all the 5 compounds do not have same solubility or cell penetration properties.

2. For figure 2D, were the copper complexes and TNF/IFN co-incubated in the primary microglia system? This data is missing the CuII(atsm) treatment and it is therefore difficult to compare any changes in MCP1 secretion between CUL5 and  CuII(atsm) treatment groups in the neuroinflammatory in vitro system.

3. The pHrodo uptake experiment is missing the CuII(atsm) treatment and it is therefore difficult to compare any changes in phagocytic activity  between CUL5 and  CuII(atsm) treatment groups under neuroinflammatory conditions. Furthermore, there should be a cytochalasin D treatment group alone without TNF/IFN.

4.  The figure 3bi is Mt1 mRNA expression but it is being to as figure 3bii in section 2.4 The CuII(atsm) treatment group is missing from these experiments.

5. The authors need to look at the Trem2 and Cd33 expression in 5XFAD treated animals in relation to activation of microglia.

6. What was the accumulation of copper complexes in the peripheral system? In an oral drug administration regimen, most of the drugs usually gets accumulated in the liver. What percentage of the drug is available after blood brain barrier penetration? Was there any change in brain tissue copper levels in the 5X FAD treated mouse. That data is crucial to compare the brain permeability of the complexes in 5XFAD mouse with respect to WT mouse.

7. The statement about mice gaining weight at 10wks of age is confusing since the treatment regimen was for 8 wks.

8. The Me-X04 experiment shows an increase in plaque phagocytosing microglia but there is an overall increase in plaque load in CuL5 treated 5XFAD mice. The authors need to address this contradiction by looking at APP expression levels and BACE activity which might also be affected by CuL5 treatment.  

9. 1.       The authors have reported an improvement in the cognitive scores in the CuL5 treated 5XFAD mice although there was increased plaque load with CuL5 treatment. Increased plaque load will eventually lead to synaptic dysfunction which is associated with cognitive deficits in AD patients and mouse models. The authors need to address this as it seems that prolonged CuL5 treatment has the potential to worsen synaptic functions and exacerbate cognitive decline.

10.Figure 5c does not appear to have a significant difference between between 5XFAD mice and CuL5 treated 5XFAD. Can the authors please confirm the statistics for the p-value significance.  

Round 2

Reviewer 3 Report

The authors have satisfactorily answered my queries. I would recommend this paper for publication with minor spelling and grammar check.

To my query number 6:

6.   Was there any change in brain tissue copper levels in the 5X FAD treated mouse. That data is crucial to compare the brain permeability of the complexes in 5XFAD mouse with respect to WT mouse. Response 6: We observed a 10% increase in brain copper content following treatment with CuL5 (Figure 4b). This was a preliminary study in which we used a mix of both WT and 5xFAD animals. 

It is not clear from the figure legend that a mixed cohort of WT and 5XFAD was used for this experiment.
